# Synthesis of Biologically Relevant 1,2,3- and 1,3,4-Triazoles: From Classical Pathway to Green Chemistry

**DOI:** 10.3390/molecules26185667

**Published:** 2021-09-18

**Authors:** Lori Gonnet, Michel Baron, Michel Baltas

**Affiliations:** 1IMT Mines Albi, UMR CNRS 5302, Centre Rapsodee, Campus Jarlard, Allée des Sciences, Université de Toulouse, CEDEX 09, 81013 Albi, France; lori.gonnet@mcgill.ca (L.G.); michel.baron@mines-albi.fr (M.B.); 2Department of Chemistry, McGill University, 801 Sherbrooke St. West, Montreal, QC H3A 0B8, Canada; 3CNRS, LCC (Laboratoire de Chimie de Coordination), Université de Toulouse, UPS, INPT, Inserm ERL 1289, 205 Route de Narbonne, BP 44099, CEDEX 4, F-31077 Toulouse, France

**Keywords:** 1,2,3-triazoles, 1,2,4-triazoles, ultrasound, medicinal chemistry, green chemistry, mechanochemistry, biological properties

## Abstract

Green Chemistry has become in the last two decades an increasing part of research interest. Nonconventional «green» sources for chemical reactions include micro-wave, mechanical mixing, visible light and ultrasound. 1,2,3-triazoles have important applications in pharmaceutical chemistry while their 1,2,4 counterparts are developed to a lesser extent. In the review presented here we will focus on synthesis of 1,2,3 and 1,2,4-triazole systems by means of classical and « green chemistry » conditions involving ultrasound chemistry and mechanochemistry. The focus will be on compounds/scaffolds that possess biological/pharmacophoric properties. Finally, we will also present the formal cycloreversion of 1,2,3-triazole compounds under mechanical forces and its potential use in biological systems.

## 1. Introduction

One of the main goals in the area of organic synthesis oriented towards biologically active compounds is the research and development of efficient environmentally safe methods. In fact, since the 2000s many regulations for the chemical and pharmaceutical industries have appeared, especially in terms of efficiency, waste management and energy input. All these issues are now addressed and termed «Green Chemistry», a multifaceted field dealing with what we call the twelve principles of P.T. Anastas and J.C. Warner [1]. Most important of them are: atom economy, preventing the use of solvents volatile and/or toxic, minimize chemical waste and minimize energy [2]. Organic reactions and processes are classically conducted in solutions (mostly organic) under reflux or thermal energy to be balanced at the end of the transformation. We focus on the Green Chemistry synthetic aspects, and focus on chemical reactions by using alternative energy sources that appeared and developed since the last two decades; namely, the processes: photochemistry through light excitation, microwave, sonochemistry irradiation, and mechanochemistry [3].

In this article, in order to give an emblematic example of the evolution of synthesis strategies towards ever greener processes, in particular in the pharmaceutical field, we will focus not only on recent classical synthesis of 1,2,3 and 1,2,4-triazoles, but also on sonochemistry and mechanochemical synthesis of these systems in relation to their biological activities. First, we will initially focus on these two alternative energy sources, i.e., sonochemistry/ultrasonic irradiation, and mechanochemistry. Mechanical effects caused by sound irradiation—called sonochemistry—can be applied to liquids. It can induce formation and growth of acoustic cavitations resulting in implosive bubble collapse [4,5,6]. This leads to intense compressional heating and extremely high pressures in the resultant so called hot spots (5000 K, 1000 atmospheres) while heating and cooling rates are exceeding 10^10^ K s^−1^ (Figure 1) [7,8]. The sonic spectrum ranges from high power to low power ultrasound (20 KHz to 10 MHz). The range from 20 KHz to 1 MHz is used in sonochemistry. As indicated before, ultrasound irradiation can induce formation and growth of acoustic cavitations resulting in implosive bubble collapse that due to their physical properties can substantially improve chemical reactions (catalytic or not) in terms of speed (in some reactions a million fold reactivity increase was observed), selectivity and yield. Ultrasound reactions are not adequate for reactions between solids or solid-gas systems [9]. Ultrasound in organic synthesis has been studied considerably in the past two decades. Especially, various named organic transformations effected through ultrasound irradiation were developed. Among the most important [10] we can point to the coupling reactions, i.e., Heck, Suzuki, Sonogashira, Ullmann, ultrasound-assisted phase transfer catalysis, some named reactions like Reformatsky, Michael, Baylis-Hillmann, but also oxidation/reduction reactions, halogenations. Finally, ultrasound synthesis of ionic liquids and heterocyclic, especially nitrogen contained compounds [11], has gained much success and development. Mechanical energy can also induce chemical transformations [10]. According to IUPAC, a mechanochemical reaction is a “Chemical reaction that is induced by the direct absorption of mechanical energy” [12,13]. Wilhelm Ostwald (Nobel Prize in 1909), was the first who mentioned the term “Mechanochemistry” and defined it as a “branch of chemistry which is concerned with chemical and physico-chemical changes of substances of all states of aggregation due to the influence of mechanical energy”. It is important to mention the pioneering work of Boldyrev et al., on the mechanisms and kinetics in comminuting devices [14,15,16], serving as a basis for many mechanochemical works. How the absorption of mechanical energy induces chemical transformations in terms of mechanistic understanding is still under investigation and not fully elucidated. Various models were proposed based on solid chemistry knowledge, like “hot spot” and “magma-plasma model” [17,18,19]. Other also well-known models (spherical, kinetic and impulse…) were equally proposed [20,21]. Many efforts were developed recently towards a mechanistic level understanding of mechanochemical processes [22]. One of the major trends in progress is to research possible links between the mechanical effect and the action of the forces generated at the molecular level [23,24,25,26,27]. In parallel with these recent advances, the topic is still subject to research from the experimental and theoretical points of view [28]. In terms of the experimental view, the traditional grinding by using a mortar and a pestle has been replaced by more sophisticated ball-milling or mechano-milling techniques that are generally conducted in vibration mills or planetary mills at frequencies of 5–60 Hz. The reactions are generally carried out in vessels or jars of different kinds of materials (stainless steel, tungsten carbide, zirconia, agate, etc.). In recent years a deviation from the pure solid status of reactants, named the liquid assisted grinding, gained considerable interest because it offers opportunities to mechanochemistry to reach viable results in comparison to solution synthesis [29,30]. These studies are greatly facilitated by a possible continuous monitoring of mechanochemical reactions [31,32]. One drawback for mechanochemistry is the fact that up to now difficulties exist in practically controlling the air and moisture sensitive reagents. However, Kubota et al. have shown recently [33] that mechanochemistry allows carrying out the syntheses of organometallics sensitive to humidity in air.

In the two last decades, this green chemistry approach has been developed considerably in areas related to inorganic compounds and metal complexes synthesis and related mechanistic aspects [34,35,36] while less interest was focused on organic mechanochemistry, even after the pioneering work reported by Toda in the 1980s [37] and Kaupp [38]. This is actually changing since the last decade’s focus was essentially on the green chemistry and green processes approach [39]. In recent times, mechanochemical synthetic approaches for creating carbon-carbon, carbon-heteroatom, metal-ligand coordination bonds etc. became important issues and gained considerable attention in the literature [40,41], and many applications were carried out in the field of organic mechanochemistry [42,43,44,45,46,47,48,49,50,51,52]. Among the many heterocyclic ring structures, especially nitrogen-contained, which were found and/or designed as important scaffolds for inducing biological effects are the triazoles. Triazole is a five membered ring with three nitrogen and two carbon atoms. Depending on the disposition of the five atoms, triazoles exist in two isomeric forms, namely 1,2,3- and 1,2,4-triazoles. Triazoles have become increasingly popular between medicinal chemists and pharmaceutical companies due essentially to their unique properties such as: rigidity, strong hydrogen-bond properties, stability under in vivo, and interesting pharmacokinetic profiles. Due to their importance, much literature data exist for 1,2,3-triazole systems in comparison to the 1,2,4-triazoles concerning either their syntheses or their biological activities see for instance [53,54,55]. In that respect the review presented here focuses on three recent parts, namely: (a)Construction of 1,2,3-triazole systems in biologically relevant compounds by means of classical and “green chemistry” conditions involving ultrasound chemistry and mechanochemistry.(b)Construction of 1,2,4-triazole systems in biologically relevant compounds by means of classical and “green chemistry” conditions involving ultrasound chemistry and mechanochemistry.(c)The mechanochemical cyclo-reversion of 1,2,3-triazole compounds and the scientific discussion on the topic that it could be extremely stimulating as mechanochemistry seems to provide a method by which reactive azide or alkyne intermediates could be selectively unmasked.

## 2. 1,2,3-Triazole Systems

One of the most important five-membered heterocyclic scaffolds due to its extensive biological activity is the 1,2,3-triazole one. The framework can be readily obtained through the click chemistry via reaction of an aryl/alkyl halide, alkynes and NaN_3_. Many synthetic methodologies were developed the past few decades, usually partitioned between metal-free and metal catalysed approaches, thus offering new opportunities for introduction of this valuable moiety to biologically relevant compounds designed and developed by medicinal chemists (Figure 1). A very recent review treats on those methodologies and on the medicinal attributes of 1,2,3-triazoles [56].

We report here notable current examples (year 2018) of classical synthesis of biologically active compounds bearing this frame, but also recent literature from 2014 concerning synthesis of 1,2,3-triazoles by chemical transformations using alternative energy sources (ultrasonic irradiation).

Alexandre et al. reported in 2018 [57] that compounds based on 4-amino-1,2,3-triazole core as potent inhibitors of indoleamine 2,3-dioxygenase (IDO1) are important targets of immuno-oncology research. The authors screened on a recombinant human IDO1 a library of 350,000 compounds and were able to identify a series bearing the 4-amino-1,2,3-triazole core. Upon chemistry optimisation they obtained compound *N*-(4-chlorophenyl)-2*H*-1,2,3-triazol-4-amine with a remarkable potency (IC_50_ of 0.023 μM) substantially more potent than any other IDO1 inhibitor. Synthesis of this compound differs from all other methods. The synthesis involves diazotization of 4-chloroaniline **1** by sodium nitrite followed by reaction with 2-aminoacetonitrile hydrochloride **2** in order to afford 2-(2-(4-chlorophenyl) iminohydrazino) acetonitrile **3** which upon heating under reflux in ethanol afforded the desired compound **4** (Figure 1).

Wu et al. [58] reported the design and synthesis of tacrine 1,2,3-triazole derivatives as potent cholinesterase inhibitors. Tacrine **5**, the first drug approved by the FDA that binds at the catalytic active site (CAS) region a potent non selective inhibitor of both bAChE and hBChE was hybridized through various types of linkers bearing the 1,2,3-triazole frame with the chloroquinoline scaffold. Starting from indoline-2,3-dione **6**, the authors obtained in a four step procedure the key tetrahydroacridine intermediates **7** bearing various terminal alkynes. After introduction of an azide functionality in the chloroquinoline scaffold **8** the partners were coupled via CuAAC reaction giving compounds **9** (Figure 2).

Among all compounds synthetized, compound **9** (R = H, linker = piperazine) exhibited a potent inhibition against AChE and BChE with IC_50_ values of 4.89 and 3.61 µM, respectively. The authors point out that although this compound is less potent than tacrine, it has a unique binding mode at both CAS and also to the peripheral anionic site (PAS), as well as less toxicity. They concluded by considering it as a lead compound that could be the basis for the development of more active dual inhibitors of AChE and BChE (Table 1).

Ashok et al. [59] reported the synthesis of a novel prototype that possessed a chromene and a 1,2,3-triazole pharmocophore frame with activities against *M. tuberculosis*. The strategy adopted by the authors for their synthesis started from substituted acetophenone **10** which upon Kabbe condensation and reduction of the carbonyl group afforded spirochromanols **11.** Deprotection of **11** and dehydration provided the corresponding spirochromene **12**. 1,2,3-triazole-fused spirochromene derivatives **13** were then obtained through a Huisgen cycloaddition in the presence of pyrrolidine as catalyst (Figure 3).

Among the compounds tested against *M. tuberculosis* H37Rv strain, 5 compounds presented strong MIC activities (between 4 and 9 μM). Their cytotoxicity against RAW 264.7 cells was determined and indicated at least one log difference in comparison to their MIC values. These findings indicated that 1,2,3-triazole-fused spirochromene derivatives can have biological significance for further development (Table 2).

López-Rojas et al. [60] reported the synthesis of 4-substituted 1,2,3-triazole coumarin-derivatives and evaluated their antimicrobial activity. The strategy adopted by the authors was the synthesis of acetylenic O- **15** or N-propargylated **17** coumarins starting from 4-hydroxy **14** and 4-bromo **16** coumarin respectively. Copper(I) catalyzed Huisgen 1,3-dipolar cycloaddition reaction with synthetized (or commercially available) alkyl or aryl azides and afforded the desired compounds **18** (Figure 4).

The authors thus created a focused library of 26 compounds with two isosteric series (hydroxy/amino) and with different substituents at the triazole moiety. Based on their MIC values against selected microorganisms, 5 out of 26 compounds showed significant antibacterial activity towards *Enterococcus faecalis* (MIC = 12.5–50 µg/mL) while low cytotoxicity was observed against human erythrocytes (Table 3).

### 2.1. Ultrasound Assisted Syntheses of 1,2,3-Triazoles

We will refer herein to some relevant publications from 2014 up to now.

In 2014, Mady et al. [61] reported the ultrasound assisted synthesis of diaryl sulfones bearing 1,2,3-triazole moieties as potential antioxidant and antimicrobial agents. Synthesis of disubstituted triazoles is depicted below (Figure 5). The authors explored three routes: a stepwise approach that allowed a click coupling of two different azides and a second and third one where both alkynes were introduced then allowed to click from the same azide. Ultrasound (US) Barbier type mediated propargylation occurred readily to construct the common intermediate **19**. This can undergo CuAAC cycloaddition reaction with different azides under ultrasound conditions affording a first 1,2,3-triazole containing compound **20**. The hydroxy group can be further propargylated and coupled with other azides (or the same), affording final compounds **23**.

The second route introduces first a second alkyne group via propargylation of the hydroxy group of key sulfone **21** and then CuAAC cycloaddition reaction with the corresponding azide. All syntheses were operated under ultrasound conditions in a very efficient manner. The authors also synthetized bis-triazoles via the one-pot click reaction (third route).

Biological and antioxidant activities of all compounds were also reported. Many of them were found to be most potent antifungal agents with MIC values around 25 μg/mL (Table 4). Moreover, compound **24** (Figure 6) showed an excellent antioxidant activity (IC_50_ = 20 µg/mL) using a DPPH free radical scavenging assay.

Nallapati et al. reported in 2015 [62] synthesis of 1,2,3-triazoles derived from olanzapine. Olanzapine (Zyprexa), a member of the thienobenzodiazepine class, is a confirmed marketed drug used for the treatment of schizophrenia and bipolar disorder. The authors describe modifications of olanzapine and explore their activities. One of the target molecules chosen by the authors being olanzapine decorated with 1,2,3-triazole moieties. In that respect alkyne **26** was first prepared through classical coupling in the presence of NaH of propargyl bromide with the drug olanzapine **25** in THF. The thus prepared alkyne reacted with aryl or alkyl azides at room temperature under ultrasound irradiation and in the presence of diisopropylethylamine affording the triazolo derivatives **27** in fairly good yields (Figure 7).

The authors reported in vitro activities of these compounds against phosphodiesterase 4B protein (PDE4B), a gene family that plays a role in the treatment of schizophrenia. Three of the compounds tested were identified as selective inhibitors of PDE4B (IC_50_ 5 to 6 μM) (Table 5).

N. Rezki reported in 2015 [63] synthesis under conventional methods and ultrasound conditions of 1,4 disubstituted 1,2,3-triazoles tethering bioactive benzothiazole nucleus and their antibacterial evaluation. Synthesis (Figure 8) started from 2-aminobenzothiazole derivatives **28** which were acylated upon reaction with bromoacetylbromide. Then, azidation in the presence of sodium azide afforded the corresponding azidobenzothiazoles **29**. All reactions were performed under classical and ultrasound conditions with better yields in the latter case. Huisgen copper(I) catalysed 1,3-dipolar cycloaddition with appropriate terminal alkynes in the presence of sodium ascorbate in *t*BuOH/H_2_O, and was carried out under heat or use of ultrasound at room temperature affording compounds **30**. Again, ultrasound conditions revealed to be more favorable.

All compounds were tested against three gram positive and three gram negative bacteria and two fungal strains. Some of them, revealed promising activities in the range of 4–8 μg/mL (Table 6).

N. Rezki and M.R. Aouad reported in 2017 [64] synthesis of hybrid compounds bearing fluorinated 1,2,4-triazole, 1*H*-1,2,3-triazole and also a benzothiazole functionality. Construction of the 1,2,4-triazole substituted frame started from reaction of 2-fluorobenzoyl chloride **31** with hydrazine hydrate.

Subsequent treatment was administered with diverse alkyl/aryl isothiocyanates, which upon basic reaction conditions underwent an oxidative ring closure affording the thione derivatives **32**. The latter reacted with propargyl bromide in the presence of triethylamine under ultrasound conditions, furnishing the thiopropargylated 1,2,4-triazole precursors **33** required for the click reaction. On the other hand, acylation of the appropriate 2-aminobenzothiazoles **34** followed by the azidolysis reaction allowed obtention of the azidoacetamide derivative **35**. The Huisgen cycloaddition reaction was then performed between the two coupling reagents in the presence of CuSO_4_ and Na-ascorbate as catalysts in DMSO-H_2_O. The ultrasound conditions were less time consuming and much more efficient with almost quantitative yields (Figure 9).

Almost all compounds showed activities with MIC values in the range 6.45–33.2 μmol/L against *S. pneumoniae*. In addition, compound **37** (Figure 2) showed the strongest antifungal activities among all compounds with MIC values of 6.45 μmol/L against *A. fumigatus* and *C. albicans*.

### 2.2. Mechanochemical Syntheses of 1,2,3-Triazoles

Praveen et al. reported in 2017 [65] the synthesis of new hybrid pharmacophores under ball milling conditions through two well established named reactions, namely a Baylis-Hillman [66] and a Huisgen’s click chemistry [67]. The authors aimed to prepare potential medicinal targets bearing a 3-substituted-3-hydroxy-2-oxindole frame present in many natural products and medicinal agents [68,69,70] and a 1,2,3-triazole scaffold. The authors successfully combined a Baylis-Hillman and a click reaction by using DABCO as a base and copper oxide nanoparticles as catalysts. By milling together a mixture of *N*-propargyl isatin, *N*-methylmaleimide, benzyl azide, DABCO in the presence of CuONP catalyst (5%) they were able to find optimal conditions of achieving the synthesis of the target compound **38a** in 96% of yield (Figure 10). 

They then applied the conditions found for creating a small focused library (Table 7) as their methodology and accommodated a large variety of substituted starting compounds. In addition, the authors proved the recyclability of the catalyst and its total recovery. Biological studies along with molecular docking demonstrated the rational efficiency of the compounds as antibacterial and antifungal. The best activities were found for compound **38m** (Figure 3), which was most active against *S. aureus* (with a MIC value of 16 µg mL^−1^), for compounds **38a.d,i,l** active against *E. coli* and for compounds **38e,h,k,m,p** active against *C. albicans*.

Sahu et al. published in 2019 [71] the synthesis of quinine-triazole systems with the aim to find new compounds via molecular hybridization [72] that can be addressed to two antiprotozoal targets that are malaria and leichmaniosis [72,73]. The synthetic route (Figure 11) started with activation through mesylation of the secondary alcohol of quinine generating the compound **39** and subsequent substitution with the azide group via a solution-based methodology [74]. The generated azido dehydroxyquinine **40** was then allowed to react via a copper catalyzed cycloaddition reaction with a variety of alkynes. These reactions were carried out under mechanochemical conditions in ball mill at 300 rpm, affording the triazolyl compounds **41** in 45% to 91% yields (Table 8). Screening results showed that from the 19 synthetized compounds, 5 showed significant antimalarial and antileichmanial activities (Table 8) and four of them did not reveal any in vivo (rodent animal model) toxic manifestation at doses as high as 1000 mg/Kg.

Finally, S. Sampath et al. reported last year [75] the synthesis of 1,2,3-triazole tethered 3-hydroxy-2-oxindoles under ball milling conditions (Figure 12) as corrosion inhibitors and antimicrobials. 3-Functionalized oxindoles can be obtained from the valuable heterocyclic scaffolds isatins. Among the different 3-substituted oxindoles, the 3-hydroxy-3-substituted-2-oxindoles are present in many natural products [76]. In addition, they are considered as valuable key intermediates in organic synthesis [77,78,79], leading to compounds with pronounced pharmaceutical properties [80,81,82]. The authors synthetized a set of new derivatives of this family by combining an aldol condensation and a click reaction using ball milling conditions. A mixture of N-propargyl isatin, acetophenone, benzyl azide in the presence of DABCO and copper oxide nanoparticles CuONPs (2.5 mol%) was reacted in a ZrO_2_ jar material at a speed of 400 rpm, affording the desired products **42** in 87% to 92% yields, except for azide possessing the strong electron withdrawing NO_2_ group (80% yield). Among the compounds synthetized, derivative **42b** displayed a remarkable corrosion inhibition potency (for corrosion inhibition in acidic media see references [83,84]), while compound **42a** showed appreciable antifungal (*C. albicans*) and antibacterial (*S. aureus*) effects (Table 9). The authors consider that the biological results are quite encouraging for triggering a detailed structure-activity study and the comprehension of their activity.

## 3. 1,2,4-Triazole Systems

The 1,2,4-triazole-based biologically active compounds have found enormous applications in medicinal and agricultural sciences. A great number of drugs are extensively used in clinics. Among them, we can point to the antifungal fluconazole **43**, antitumoral letrozole **44**, and the antiviral ribavirin **45**, (Figure 4) [85], while several triazole based compounds play an important role in agriculture ensuring harvest and crops [86]. Their extensive medicinal, agrochemical potential, resulted in an overwhelming effort to develop synthetic methods that include three categories of synthetic objectives: (a) cyclizations to form the triazole ring, (b) transformations of heterocyclic compounds to construct the triazole ring, and (c) substitutions on the 1,2,4-triazole ring.

In this review, we discuss only some examples of the recent cyclization reactions with amidrazones and hydrazides. In addition, it is noteworthy to point out that there are no reported methods to synthetize 1,2,4-triazole frames under green chemistry conditions. We report here our first results concerning the mechanochemical organic synthesis of a valuable annulated 1,2,4-triazole scaffold.

### 3.1. From Amidrazones

Amidrazones are the conjugated products of imines and hydrazines; their cyclisation with carbonyl compounds is one of the most important pathways to access 1,2,4-triazole derivatives. 

In 2015, Nakka et al. [87] reported an environmentally benign protocol for the synthesis of 3,4,5-trisubstituted 1,2,4-triazoles **48**. The authors performed the coupling/cyclisation reaction by heating in polyethylene glycol and in the presence of ceric ammonium nitrate (catalyst, 5%), *N*-arylamidrazones **46** and aldehydes **47**. The authors demonstrated that this protocol could generate good yields of 3,4,5-tri-substituted 1,2,4-triazoles bearing different functionalities, while in addition, the effective recyclability of the medium could make the process industrially interesting (Figure 13). 

Szőcs et al. [87] have already shown that 1,2,4-triazole frames judiciously attached at the 5-position of a 3-*C*-glucopyranosyl scaffold give access to very efficient inhibitors of glycogen phosphorylase. In that respect they become extremely important hits as potential antidiabetic agents (type 2 diabetes). Szőcs et al. reported in 2015 [88] 38 new developments concerning the synthetic approaches for this class of compounds and in particular the oxidative ring closures of *N*^1^-alkylidene carboxamidrazones. When glycosyl cyanides **49** and amidrazones **50** were treated under Raney reductive conditions in the presence of NaHP_2_O_2_, they afforded the two tautomeric forms **51** of O-peracylated *N*^1^-(β-*D*-glycopyranosyl-methylidene)-arenecarboxamidrazones. Bromination of **52** by *N*-bromosuccinimide (NBS) led to halogenated **47** type derivatives. The latter can then undergo in pyridine or by NH_4_OAc in AcOH, a ring closing reaction to the desired 3-*C*-glycosyl-5-substituted-1,2,4-triazoles **53** (Figure 14). 

The authors took advantage of their methodology to create 3,5-diaryl-1,2,4-triazoles **55** starting from *N*^1^-arylidene-arenecarboxamidrazones **54**. Reaction of the latter with NBS/NH_4_OAc in AcOH (whatever the order) afforded triazoles **55**, thus demonstrating the general applicability of the method (Figure 15). 

Among the different compounds synthetized and tested, it is noteworthy to point out that compound **56** (Figure 5) with an inhibition constant Ki of 0.41 μM against rabbit muscle glycogen phosphorylase could be considered as the starting point for the development of more potent compounds for pharmacological treatment, not only of diabetes but also wherever the regulation of glycogen metabolism plays a significant role (cerebral and cardiac ischemias, and tumor growth).

In 2018, Aly et al. [89] reported a general method for the synthesis of 1,3,5-trisubstituted 1,2,4-triazoles **59**, **60** from reaction of amidrazones **57** with diethyl azodicarboxylate **58**. The authors performed the coupling/cyclisation reaction between *N*-arylamidrazones **57** and diethyl azodicarboxylate **58**. The reaction was conducted under reflux in EtOH and catalyzed by a few drops of triethylamine (Figure 16), thus allowing to get an easy access to the highly diverse triazoles **59**, **60**. The reaction is based on oxidation of ethanol to acetaldehyde via the Mitsunobu reagent; upon reaction with amidrazones the substituted 3-methyltriazoles **59** could be obtained, while a [2  +  3] cycloaddition reaction between two oxidized forms of amidrazones produced triazoles **60**. 

### 3.2. From Hydrazides

Several hydrazides are commercially available and the non-commercial ones are successfully prepared by the reaction of hydrazine with the corresponding ester precursor. Up to now, a lot of works have been done concerning the cyclization of hydrazides or their derivatives. Very recent work in relation to biological activities is presented here.

In 2018, Singh et al. [90] reported the design and synthesis of new bioactive 1,2,4-triazoles as potential antituberculosis and antimicrobial agents. In that respect, they synthetized a series of functionalized 1,2,4-triazole derivatives through the synthetic scheme presented below (Figure 17).

Isonicotinic acid hydrazide **61** was transformed to the potassium dithiocarbazinate **62** by reaction with carbon disulfide under basic conditions. Then, treatment with hydrazine hydrate under thermal conditions in water, afforded the 4-amino-1,2,4-triazole-3-thiol **63**. The latter reacted on its 4-amino group to form various Schiff base compounds. Some of the compounds were found to have very potent antitubercular activities, even better than isoniazid and also against clinical isolates (Table 10). 

Synthetized compounds were also tested in vitro against representative bacterial and fungi strains, one compound has very potent activity against *B.subtilis*, while another one is very potent against *A.niger* and *C. albicans* fungi (Table 11).

Sonawane et al. reported in 2017 [91] the synthesis of 1,2,4-triazole-3-thione derivatives as antimycobacterial agents. The two routes employed by the authors are outlined in Figure 18. The acid chloride **66**, prepared by reacting aromatic carboxylic acid **61** with thionyl chloride reacted with thiosemicarbazide, which without isolation and upon thermal heating under aqueous basic conditions, led to the desired compounds **67** (route A). Triazolethiones **69** could not be synthetized by this procedure were obtained by reaction of hydrazide **68** with carbon disulfide, followed by heating in the presence of a 25% ammonia solution (route B). 

Two of the compounds synthetized showed high antitubercular activity against the dormant H37Ra strain in vitro and ex vivo; they also showed extremely low cytotoxicity and high solubility indicating the potential of developing these compounds further as novel therapeutics against tuberculosis infection (Table 12).

Liu et al. [92] reported in 2017 a family of 7-hydroxy-4-phenylchromen-2-linked 1,2,4-triazoles with potent antitumoral activities. The synthetic procedure adopted made use of the coumarin synthetized derivatives **70** that were functionalized by reaction with ethylbromoacetate followed by transformation of the ester group to a hydrazide functionality. The latter, when condensed with dimethylacetal followed by a strong thermal reaction with an amine in the presence of glacial acetic acid afforded the triazole derivatives **73** in good overall yields (Figure 19).

The new 1,2,3-triazole derivatives showed improved antiproliferative activities. Particularly, compound **74** exhibited potent activity with important IC_50_ values against AGS (2.63 ± 0.17), MGC-803 (3.05 ± 0.29) and HCT-116 cell lines (11.57 ± 0.53 µM) (Figure 6). The authors also demonstrated that these compounds had strong activity against the HeLa cell line, with an IC_50_ value of 13.62 ± 0.86 µM. All activities were better than those of the non-substituted 7-hydroxy-4-phenyl-2*H*-chromen-2-one **70** (with R = H) and that of the positive control drug 5-fluorouracil. Moreover, the authors showed that the biological activities of the 1,2,4-triazole derivatives were significantly higher than that of the 1,2,3-triazole ones. 

### 3.3. Annulated 1,2,4-Triazole Systems

Among the annulated 1,2,4-triazole systems we will present the triazolophthalazine frame that was developed in our group by conventional and non-conventional means.

Some years ago, De et al. [93] explored the possibility of cinnamic acid derivatives as potential antituberculosis agents. In the course of their studies the authors synthetized 4-alkoxy cinnamoyl derivatives resulting from the coupling of the corresponding acids (or activated ones) with different nucleophiles and among them amines, hydrazines, thiols.

In the course of their first studies, when reacting under peptide coupling conditions (EDC, HCl, HOBt, and trimethylamine), 1-hydrazinophthalazine hydrochloride **75**, and cinnamic acid derivatives for 48 h under reflux in acetonitrile, the authors obtained in good yields the corresponding 3-(4-alkoxystyryl)-[1,2,4]triazolo[3,4-a]phthalazines **77** (65–90%). This was formed through a coupling-intramolecular cyclization-dehydration sequence (Figure 20).

All alkoxylated compounds showed good antitubercular activities. More importantly, triazolophthalazine derivative **73** (Figure 7), bearing a 4-isopentenyloxy chain on the phenyl ring, showed excellent antitubercular potency (MIC = 1.4 μM) in addition to a very good cytotoxicity toward HCT116 human cells (IC_50_ = 449 μM; 160 μg/mL and selectivity index SI = 320). It is also noteworthy to point out that this compound does not act on the mycolic acid biosynthesis of mycobacteria and up to now its target is unknown.

In order to build a small, focused library of styryltriazolophthalazines, Veau et al. [94] modified the convergent route to a divergent one, by exploring the possibility of the construction of the phenolic precursor **79** that could then lead to various alkoxylated derivatives of type **77** (Figure 21).

The target precursor **79** was thus obtained by the authors in a two step procedure: 1-hydrazinophthalazine hydrochloride **74** and *p*-hydroxy cinnamic acid **80** when reacted in acetonitrile for 1 h under peptidic coupling conditions but under microwave afforded a 69% yield after a simple filtration–recrystallization sequence of the styrylphenolic triazolophthalazine **79**. Alkylation by various alkylating agents and under standard conditions led to the desired alkoxylated derivatives **81** (alkylating agent, K_2_CO_3_, KI, DMF, 60 °C, overnight) (Figure 22).

In continuation of this work on the triazolophtalazine frame the authors considered that this could be an interesting pharmacophore to explore. In that respect they also synthetized two compounds bearing either an alkyne group or a bromine at the 2 position of the 1,2,4-triazole frame.

Concerning the alkyne compound, after several attempts, Veau et al. [94] considered the best way to obtain it is a two-step sequence. Coupling of trimethylsilyl propiolic acid **82** with 1-hydrazinylphthalazine hydrochloride **74** under mild conditions afforded acyclic precursor **83**. Under microwave irradiation in acetonitrile for 1 h, the precursor was transformed quantitatively to the cyclized silylated intermediate **84**. Upon standard deprotection conditions (i.e., K_2_CO_3_ in MeOH) compound **84** led to derivative **85** (31% overall yield over three steps) (Figure 23).

For the first time, Gonnet et al. [95] reported in 2019 the mechanochemical synthesis of 1,2,4-triazoles starting from hydralazine hydrochloride. By using a planetary ball-mill, and in the presence of pyrogenic S13 silica as the grinding auxiliary, total conversion to intermediate hydrazones **86** was achieved in a few minutes (Figure 24). Iodobenzene diacetate (IBD) was used for optimal conversion of nonphenolic hydrazones to annulated 1,2,4-triazoles **87**, while SeO_2_ was found to be efficient for phenolic compounds (Figure 25).

In addition, for the first time, the one-pot two-step synthesis (Figure 24) leading to annulated 1,2,4-triazoles was also successfully conducted.

Comparison to the conventional syntheses of hydrazone **86** and triazole **87** clearly showed the green metrics overall efficiency of the mechanochemical synthesis (Table 13).

Synthesis of the brominated compound **91** was also explored. First a conventional method was performed by Veau et al. [94], by reacting overnight under reflux in ethanol, 1-hydrazinyl-phthalazine hydrochloride **74** with trimethylortho-formate in the presence of some drops of acetic acid in order to obtain unsubstituted triazolophthalazine **90**. Then, reaction with bromine in the presence of pure acetic acid under reflux afforded the brominated compound **91** in 53% yield.

More recently, Gonnet et al. [95] reported the two-step bromination reaction successfully conducted by mechanochemistry. Reaction of 1-hydrazinylphthalazine hydrochloride **74** in the presence of trimethylorthoformate and some drops of acetic acid reacted in a planetary ball mill for 1 h affording quantitatively triazolo-phthalazine **90**. Reaction in the same planetary ball-mill (PBM) of the triazolophthalazine with sodium bromide, oxone and some silica afforded after 1 h the brominated compound **91** in quantitative yield (Figure 26).

All compounds bearing the 1,2,4-triazole frame were evaluated for various biological properties. The 3-aryl substituted 1,2,4-triazole derivatives **87** and **89** do not present valuable activities against *M.tuberculosis* ((MIC around 80 μM) [96]. The alkyne derivative **85** present a very good activity (MIC 12.9 μM) while the brominated **91** is much less potent (MIC 40 μM) [94]. The alkyne derivative **85** did not manifest cytotoxicity toward HCT116 human cells while it is equally active against multidrug-resistant *M. tuberculosis* strains. Considering all results starting from compound **73**, it seems likely that the triazolophthalazine could be an important scaffold in order to obtain new families of compounds with strong antitubercular activity and an alternative mode of action for compared with standard anti *M. tuberculosis* drugs.

In that respect, the authors consider the possibility of developing focused libraries of triazolophthalazine compounds by using the two important precursors that are the alkyne and the brominated derivatives and developing coupling reactions under conventional and/or mechanochemical means. In addition, further work is necessary to tackle the identification of the protein targeted by this class of potent anti *M. tuberculosis* compounds.

## 4. Mechanochemical Cycloreversion of 1,2,3-Triazoles

Globally, the cycloaddition process is strongly favored thermodynamically (ΔH = −45 to −55 Kcal/mol) [97]. The 1,2,3-triazole frame is robust and inert under most thermal chemical treatments but also in aqueous or biological environments. In 2011, Brantley et al. [98] reported the possibility of unclicking the click on specific 1,4-substituted 1,2,3-triazoles by mechanical forces. They first hypothesized that mechanical exogenous forces directed to judiciously chosen scaffolds incorporated in a polymer chain can formally disallow pericyclic reactions. The authors incorporated the triazole ring in polymer chains and one of them was judiciously chosen, when mechanical ultrasound forces were applied (ultrasonication in a Suslick cell at 0 °C) resulted in the cleavage of the triazole ring **92** to its alkyne **93** and azide **94** components (Figure 27).

They concluded that the ability to selectively deconstruct triazoles might serve to elaborate mechano-responsive materials for potential controlled bioconjugation applications or force responsive fluorescent tags for biological assays.

This stimulated a puzzling publication (that was retracted since by the editor) [99] aroused strong debate. The same authors, based on sonochemical experiments related to extended Bel theory, discussed and concluded on the lowering of the activation energy barrier for cycloreversion [100] through application of an external force to the triazole ring. In the contrary, purely theoretical work, it was shown that the cycloreversion barrier is as high as 70 Kcal/mol [101]. In addition, the mechanochemically induced retro-click of the 1,2,3-triazole ring vs. bond rupture next to it could not be unambiguously concluded when single molecule force spectroscopy experiments were applied [100]. Stauch and Dreuw reported in 2017 [102] a theoretical work where by using the JEDI (Judgment of Energy DIstribution) analysis it was concluded that for 1,4 disubstituted triazoles the unclick reaction is impossible, even when Cu^I^ assisted (Figure 28a). For 1,5-disubstituted triazoles where a parallel alignment of the scissile bond exists, this could be feasible. Nevertheless, the retro click cycloreversion is not selective as it competes with the carbon-nitrogen bond connecting the triazole ring to the linker. During the same year, Krupička et al. [103] also concluded that only in these 1,5-disubstituted 1,2,3-triazole systems are the Gibbs free energy barriers 55 Kcal/mol (unclick reaction) versus 45 Kcal/mol for external C-N bond cleavage. The authors also point out an extremely exciting finding by showing that the calculated Ru-assisted mechanochemical cycloreversion of the 1,5 regioisomer dramatically lowers the activation energy of the rate determining step down to 20 Kcal/mol (the first step), while the decomplexation of the cleaved intermediate readily occurs, leading to the alkyne and azide components (Figure 28b).

In conclusion, the authors point out that the Ru-assisted mechanochemical unclicking of the 1,5 regioisomer could be an extremely selective process. If this is to be experimentally proved it would open the path for very important potential applications.

## 5. Conclusions

Among the nitrogen-contained heterocyclic ring structures one of the most important providing long term advancement in the medical field are triazoles. They became in the last decades the heterocycle of choice in all fields of drug discovery receiving much of the attention and offering new opportunities for medicinal chemists.

We must point out that while 1,2,3-triazole systems are very well documented in terms of classical organic synthesis, their synthetic methodologies under green chemistry approaches based on less energy input requirements are beginning to emerge, but are still focused on ultrasound reactions. It is our feeling that the mechanical approaches will be further developed (mechanochemical synthesis, ultrasound). There is still a lot to be invented and this is a great opportunity for the chemists and medicinal chemists, but also in extenso for the pharmaceutical industry.

The same and in a greater extent is also true for the synthesis of the valuable regioisomeric scaffold of 1,2,4-triazole systems. Except for our contribution in the field, there is no other green chemistry (ultrasound or mechanochemistry) developed for this family of compounds. We believe that here also great opportunities exist for all communities of synthetic, physical, theoretical and medicinal chemists, whether they are in the academia or in the industry.

Finally, concerning the mechanochemical (ultrasound) unclicking of the 1,5 disubstituted 1,2,3-triazole, active experimental research work is needed in order to create an extremely selective process that could also confirm the theoretical work. That could pave the way for important biological and other applications.

## Data Availability

Not applicable.

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
