# Peer review of "Synthesis of Biologically Relevant 1,2,3- and 1,3,4-Triazoles: From Classical Pathway to Green Chemistry"

_molecules, 2021, doi:10.3390/molecules26185667_

Round 1
Reviewer 1 Report
This manuscript was submitted for a special issue on mechanochemistry, which is an important and underappreciated sub-discipline of chemical synthesis. The authors are recognized for their contributions in this area, and as such are well positioned to review the field.
The review itself, however, needs to be more focused. In particular, the inclusion of extensive biological activity is distracting and incidental at best to the central message. The authors may have chosen to limit the scope of their review to triazoles prepared in the context of medicinal chemistry efforts, but this choice is problematic. The central concepts and virtues of mechanochemistry transcend the properties of the product triazole; the review would be more impactful if it included more examples of applications to triazole synthesis, including triazoles that were subsequently examined for biological activity. Furthermore, the tables of biological activity data are out of context and distracting to the central message. The triazole may be a common subunit among the compounds highlighted here for biological activity, but the triazole is not a key, unifying pharmacophore (any more so than is, for example, the pyrazine ring, or other small heterocycle). The authors may choose to reference the biological activity of various triazoles to underscore that triazoles (like other heterocycles) are often made in the context of medicinal chemistry efforts, but a review of the data themselves should not be part of this review on non-conventional strategies and tactics for chemical synthesis.
Publication is not recommended at this time. The authors should remove the pharmacology data from their review and focus on the chemical synthesis. It may be fine to limit the scope of the review to biologically relevant triazoles, provided that there is nothing more to be gained (learned) from reviewing triazoles made in other contexts. On the other hand, if there are instructive examples of non-conventional methods of synthesis applied recently to the synthesis of triazoles in other contexts (including pure synthetic methods development, which are often unbiased in the potential applications of the products), then these should be included in the review as well.
In summary, I recommend reconsideration after major revisions. The pharmacology data should be removed (it would still be appropriate to cite the studies and refer to the activity: e.g., "triazole 9 shows promise as a potential lead compound for dual inhibition of AChE and BChE [59]."), and the authors should consider including any instructive examples of triazoles that do not fall under the (somewhat arbitrary) umbrella of having been subsequently examined for biological activity.
Author Response
Manuscript ID molecules-1350895
Responses to the Reviewers
The the authors are particularly grateful to the Reviewers who carried out a significant investigation of the submitted manuscript, investing precious time in this task. Responses to comments and suggestions are listed below. Responses to Reviewer 1. We thank the Reviewer 1 for his overall analysis of the manuscript through his prism, and we will give him the answers to his impressions and questions.
Reviewer 1. This manuscript was submitted for a special issue on mechanochemistry, which is an important and underappreciated sub-discipline of chemical synthesis. The authors are recognized for their contributions in this area,and as such are well positioned to review the field.
Response 1.The authors totally agree with this, and acknowledge Reviewer 1 for his very positive assessment, for considering them perfectly legitimate to carry out a review in this field.
Reviewer 1. The review itself, however, needs to be more focused. In particular, the inclusion of extensive biological activity is distracting and incidental at best to the central message. The authors may have chosen to limit the scope of their review to triazoles prepared in the context of medicinal chemistry efforts, but this choice is problematic. The central concepts and virtues of mechanochemistry transcend the properties of the product triazole; the review would be more impactful if it included more examples of applications to triazole synthesis, including triazoles that were subsequently examined for biological activity. Furthermore, the tables of biological activity data are out of context and distracting to the central message. The triazole may be a common subunit among the compounds highlighted here for biological activity, but the triazole is not a key, unifying pharmacophore (any more so than is, for example, the pyrazine ring, or other small heterocycle). The authors may choose to reference the biological activity of various triazoles to underscore that triazoles (like other heterocycles) are often made in the context of medicinal chemistry efforts, but a review of the data themselves should not be part of this review on nonconventional strategies and tactics for chemical synthesis.
Response 2.
The authors share the Reviewer’s opinion that central concepts and virtues of mechanochemistry transcend the properties of the product triazole. On other points, the reviewer’s opinion is perfectly respectable, but different from ours. 75% of drugs contain nitrogen heterocycles, and this proportion is constantly increasing. The choice of the authors is focused on the synthesis of 1,2,3- and 1,3,4- triazole nuclei, for which they have in-depth skills, both in synthesis and in mechanochemistry. This manuscript is not intended to be a compilation of nitrogen heterocycles synthetic processes as it had already been done. On the contrary, it aims to show, through an emblematic example of nitrogenous pharmacophore, the resistible progress of green chemistry, in the field of therapeutic chemistry which has recently opened up to sustainable processes. Since a majority of the progress in the field is due to research carried out on potential drugs, this explains the choice of the title. The manuscript does not exclude that syntheses be carried out with a view to other applications. By way of example, compound 42b of the manuscript was synthesized for corrosion inhibition applications.
The literature search had in fact initially been carried out as a broad sweep of these syntheses. But it was clear that the vast majority of the products sought and obtained had a biological activity, and had been synthesized within the framework of medicinal chemistry, hence the choice of the title. Hence also the fact that the tables of biological data are located as close as possible to the syntheses of products, to give those skilled in the art a rapid answer to their questions on the biological activity obtained. A general picture would have been too indigestible and inappropriate in the eyes of the authors.Despite this difference of opinion on the form of the manuscript, we sincerely thank the reviewer for his wise advice, and most respectable. The text of the manuscript has been improved to explain the approach used and the objective of the review. A sentence has been introduced to present the objective more clearly: “In this article, in order to give an emblematic example of the evolution of synthesis strategies towards ever greener processes, in particular in the pharmaceutical field, we will focus besides recent classical synthesis of 1,2,3 and 1,2,4 triazoles”
Reviewer 1. Publication is not recommended at this time. The authors should remove the pharmacology data from their review and focus on the chemical synthesis. It may be fine to limit the scope of the review to biologically relevant triazoles, provided that there is nothing more to be gained (learned) from reviewing triazoles made in other contexts. On the other hand, if there are instructive examples of non-conventional methods of synthesis applied recently to the synthesis of triazoles in other contexts (including pure synthetic methods development, which are often unbiased in the potential applications of the products), then these should be included in the review as well.
Response 3.As explained in response 3, the reader involved in therapeutic chemistry naturally expects to be able to consult the pharmacological data close to the synthetic routes. The text already includes example (compound 42b - Sampath 2020) of activity other than a biological activity (corrosion inhibition), which shows that the syntheses selected in the manuscript are of general interest, and by nature generic. After in-depth study of the literature in order to update innovative synthesis strategies, some references were added as presented in response 5.
Reviewer 1. In summary, I recommend reconsideration after major revisions. The pharmacology data should be removed (it would still be appropriate to cite the studies and refer to the activity: e.g., "triazole 9 shows promise as a potential lead compound for dual inhibition of AChE and BChE [59]."), and the authors should consider including any instructive examples of triazoles that do not fall under the (somewhat arbitrary) umbrella of having been subsequently examined for biological activity.
Response 4.At the request of Reviewer 1, the manuscript was revised as much as possible, but without distorting or destructuring the whole text, which has already been carefully designed and refined by the authors. To go inline with the Reviewer’s recommendations, some recent references were introduced [53-55], relating to triazole synthesis strategies and tactics. As noted above (responses 3 & 4), these syntheses are mainly oriented towards obtaining medicinal chemistry products. The authors are very grateful to the Reviewer 1 for his suggestions, which allowed to deepen the reflexion around the manuscript, and consequently to significantly improve the meaning and the content of the original manuscript.
Reviewer 2 Report
The review article submitted by Michel Baltas and coworkers explain the synthesis of pharmacologically relevant triazoles. The review is interesting and can be accepted for publication after addressing the following concerns:
1) Page 1, line 15, Abstract: Change "light, ultrasound" to "light and ultrasound".
2) In keywords, "medicinal green chemistry" seems to be irrelevant. "Medicinal chemistry" and "Green chemistry" would be good.
3) Page 1, line 27: Restructure the last part of the sentence starting from "research, development of efficient and environmentally safe methods".
4) Page 2, line 89: The last part "sensitive organometallic in air" is not understandable. Please change it to a clear readable form.
5) Page 4, line 139: Italicize "H" in N-(4-chlorophenyl)-2H-1,2,3-triazol-4-amine.
6) Page 6, line 213: Change "1,2,3-triazoles moieties" to "1,2,3-triazole moieties".
7) Page 6, line 223: The phrase "21 then" is not clear. Better to show as "21 and then".
8) Page 7, line 233: Change to "Moreover, the compound 24".
9) Page 8, line 268: Italicize "t" in tBuOH.
10) Page 9, line 284: Italicize "H" in "1H-1,2,3-triazole".
11) Page 10, line 298: Change "yields almost quantitative" to " almost quantitative yields".
12) Page 10, line 311: Change "the synthesis under ball milling conditions of" to "the synthesis of new hybrid pharmacophores under ball milling conditions through....".
13) Page 11, line 335: Re-frame the sentence starting from "The best activities were found for....".
14) Page 11, line 347: Change "allowed then" to "then allowed".
15) Page 12, line 362: Change "antimicrobial ones" to "antimicrobials".
16) Page 12, line 370 and 371: Change "2" to subscript for "ZrO2" and "NO2".
17) Page 14, line 436: Change "N1-Arylidene-arene" to "N1-arylidene-arene".
18) Page 15, line 464, Scheme 16: Change "amidrazones 56" to "amidrazones 57".
19) Page 17, line 511: Change "showed also" to "also showed".
20) Page 17, Table 12: "Anti-tubercular activity against MTB (μL/mL)". Is it "(μg/mL)"?
21) Page 18, line 545: Correct to "triazolophthalazine".
22) Page 19, line 584: Correct "79,." to "79.".
23) Page 21, line 633 and 640: Correct "trimethylorthoformate".
24) Page 21, line 670: Correct "biologically environments" to "biological environments".
25) The authors should cite some recent references for the diverse synthesis of biologically relevant 1,2,3 and 1,2,4-triazoles, For example: a) Journal of the Mexican Chemical Society, 2020, 64 (1), 53-73; (b) Journal of Heterocyclic Chemistry, 2020, 57 (8), 3173-3185; (c) Journal of Chemical Sciences, 2021, 133 (1), 1-12.
26) Apart from these, all the schemes should be checked again for proper subscripts for formula and wordings. For instance there are mistakes or typos in Schemes 2, 5, 8, 11, 12, 15 and 26.
After addressing all these comments, the review article may be published.
Author Response
Manuscript ID molecules-1350895
Responses to reviewers
The authors are particularly grateful to the reviewers who carried out a significant investigation of the submitted manuscript, investing precious time in this task. Responses to comments and suggestions are listed below. Responses to Reviewer 2. We thank the Reviewer 2 for his very detailed and particularly relevant analysis of the manuscript. The Reviewer’s comments have been taken into account, and the responses are shown below.
1) Page 1, line 15, Abstract: Change "light, ultrasound" to"light and ultrasound".
Response 1. The requested modification has been carried out.
2) In keywords, "medicinal green chemistry" seems to be irrelevant. "Medicinal chemistry" and "Green chemistry" would be good.
Response 2. The requested changes have been made. 3) Page 1, line 27: Restructure the last part of the sentence starting from "research, development of efficient and environmentally safe methods". Response 3. The sentence was replaced by : research and development of efficient environmentally safe methods.
4) Page 2, line 89: The last part "sensitive organometallic in air" is not understandable. Please change it to a clear readable form
Response 4. The sentence was clarified : “the syntheses of organometallics sensitive to humidity in air.”
5) Page 4, line 139: Italicize "H" in N-(4-chlorophenyl)-2H-1,2,3-triazol-4-amine.
Response 5. The requested modification has been carried out.
6) Page 6, line 213: Change "1,2,3-triazoles moieties" to "1,2,3-triazole moieties".
Response 6. The requested modification has been made.
7) Page 6, line 223: The phrase "21 then" is not clear. Better to show as "21 and then".
Response 7. The requested modification has been made.
8) Page 7, line 233: Change to "Moreover, the compound 24".
Response 8. The requested modification has been made. 9) Page 8, line 268: Italicize "t" in tBuOH.Response 9. The requested modification has been made 10) Page 9, line 284: Italicize "H" in "1H-1,2,3-triazole".Response 10. The requested modification has been made
11) Page 10, line 298: Change "yields almost quantitative" to " almost quantitative yields".
Response 11. The requested modification has been made.
12) Page 10, line 311: Change "the synthesis under ball milling conditions of" to "the synthesis of new hybrid pharmacophores under ball milling conditions through....".
Response 12. The requested modification has been made
13) Page 11, line 335: Re-frame the sentence starting from "The best activities were found for....".
Response 13. The sentence was replaced by: The best activities were found for compound 38m (Fig. 5a) most active against S.aureus (with a MIC value of 16µg mL-1), for compounds 38a.d,i,l active against E.coli and for compounds 38e,h,k,m,p active against C.albicans.
14) Page 11, line 347: Change "allowed then" to "then allowed".
Response 14. The requested modification has been made
15) Page 12, line 362: Change "antimicrobial ones" to "antimicrobials".
Response 15. The requested modification has been made
16) Page 12, line 370 and 371: Change "2" to subscript for "ZrO2" and "NO2".
Response 16. The requested modification has been made
17) Page 14, line 436: Change "N1-Arylidene-arene" to "N1-arylidene-arene".
Response 17. The requested modification has been made
18) Page 15, line 464, Scheme 16: Change "amidrazones 56" to "amidrazones 57".
Response 18. The requested modification has been made
19) Page 17, line 511: Change "showed also" to "also showed".
Response 19. The requested modification has been made
20) Page 17, Table 12: "Anti-tubercular activity against MTB (μL/mL)". Is it "(μg/mL)"?
Response 20. The requested modification has been made 21) Page 18, line 545: Correct to "triazolophthalazine".Response 21. The requested modification has been made 22) Page 19, line 584: Correct "79,." to "79.".Response 22. The requested modification has been made
23) Page 21, line 633 and 640: Correct "trimethylorthoformate".
Response 23. The requested modification has been made
24) Page 21, line 670: Correct "biologically environments" to "biological environments".
Response 24. The requested modification has been made
25) The authors should cite some recent references for the diverse synthesis of biologically relevant 1,2,3 and 1,2,4-triazoles, For example: a) Journal of the Mexican Chemical Society, 2020, 64 (1), 53-73; (b) Journal of Heterocyclic Chemistry, 2020, 57 (8), 3173-3185; (c) Journal of Chemical Sciences, 2021, 133 (1), 1-12.
Response 25. The references were rationalized and the suggested references were added.
26) Apart from these, all the schemes should be checked again for proper subscripts for formula and wordings. For instance there are mistakes or typos in Schemes 2, 5, 8, 11,12, 15 and 26.
Response 26. The requested changes have been made.
After addressing all these comments, the review article may be published
Response. The authors are much grateful to the Reviewer 2 for the constructive and very positive proposals and totally agree with the changes recommended. Thanks to the recommendations, the manuscrit was totally revised and greatly improved. Furthermore, small typographic details were improved, mainly in the references.
Round 2
Reviewer 1 Report
Not much has changed from the original manuscript, and I still think the pharmacology data is incidental and distracting. However, I agree that there is enough value in the mechanochemistry information to merit publication in Molecules.